# Profile Likelihoods on ML-Steroids

Theo Heimel[1,2], Tilman Plehn[1,3], and Nikita Schmal[1]

**1** Institut für Theoretische Physik, Universität Heidelberg, Germany
**2** CP3, Université catholique de Louvain, Louvain-la-Neuve, Belgium
**3** Interdisciplinary Center for Scientific Computing (IWR), Universität Heidelberg, Germany

November 5, 2024

## Abstract

**Profile likelihoods, for instance, describing global SMEFT analyses at the LHC are numerically expensive to construct and evaluate. Especially profiled likelihoods are notoriously unstable and noisy. We show how modern numerical tools, similar to neural importance sampling, lead to a huge numerical improvement and allow us to evaluate the complete SFitter SMEFT likelihood in five hours on a single GPU.**

# 1 Introduction

With the shift of the main physics paradigm of the LHC towards data-driven and bottom-up precision analyses, global SMEFT analyses allow us to answer the key question *Does the LHC data agree with the Standard Model altogether?* The reason is that SMEFT covers all sectors of the Standard Model and allows us to combine huge numbers of rate and kinematic measurements in a theoretically meaningful way. These analyses define the Run2 legacy in the Higgs-gauge sector [1–4], the top sector [5–9], both sectors combined [10–12].

The problem with global analyses, and the likely reason why none of the LHC experiments have published a proper global analysis, is that sampling the likelihood over the common space of Wilson coefficients and nuisance parameters is extremely CPU-intensive, especially when we want to avoid simplified Gaussian distributions. Removing nuisance parameters and unwanted Wilson coefficients through profiling, rather than the numerically easier marginalization, adds to the numerical misery.

The first way to improve the likelihood evaluation is to use the structure of SMEFT as a perturbative theory to simplify this sampling problem significantly. Secondly, modern machine learning and automatic differentiation allows us to sample the likelihood in a parallelized and efficient manner. Finally, neural importance sampling [13–16], for instance implemented in MadNIS [15, 17, 18], allows us to sample the likelihood much more efficiently.

In this paper we study for the first time how a SFITTER global SMEFT analysis can be accelerated by combining these three directions. In Sec. 2 we review the construction of the SFITTER likelihood and its perturbative structure and propose five steps to numerical happiness. In Sec. 3 we briefly summarize the construction of the SFITTER likelihood, such that we illustrate the five steps for a simplified toy model in Sec. 4.1 and then show that they also work for the full SMEFT analysis in Sec. 4.2. For the first time, this new method allows us to extract all Wilson coefficients for the Higgs-gauge and top sectors including the full correlated set of uncertainties in a numerically stable manner. In general, our numerical improvements allow us to compute SMEFT profile likelihoods in a few hours on a single GPU, rather than on a CPU cluster over days and weeks.

# 2 The five steps to happiness

The physics task behind our improved likelihood sampling is a global SMEFT analysis, combining the top and Higgs-gauge sectors including a comprehensive uncertainty treatment. With established methods, this analysis is possibly, but extremely CPU-intensive. In Sec. 2.1 we will describe the structure of the fully exclusive likelihood and the challenges in evaluating it fast. In Sec. 2.2 we will introduce the five steps to sampling happiness.

## 2.1 Constructing the likelihood

For decades, the exclusive likelihood for a single measurement as a function of model parameters $c$ in SFITTER has been constructed as [19]

$$L_{\text{excl}}(\theta) = \text{Poiss}(d|p(c, \theta, b)) \, \text{Poiss}(b_{CR}|bk) \prod_i C_i(\theta_i, \sigma_i) \, . \tag{1}$$

It includes the Poisson probability to observe $d$ events with $p$ events predicted, where the prediction is affected by uncertainties encoded in nuisance parameters $\theta$. The backgrounds $b$ are determined by an event count $b_{CR}$ in the control region and an appropriate interpolation $k$

to the signal region. The constraints $C_i$ describe the distributions of the nuisance parameters $\theta_i$ with corresponding width $\sigma_i$.

**Top sector**

For the top fit [7, 12] we use signal strengths instead of rate measurements. Each measurement is then modeled as a Gaussian rather than a Poisson distribution. For unfolded data, backgrounds do not need to be taken into account, so the likelihood becomes a product of Gaussians,

$$L_{\text{excl}}(\theta) = \mathcal{N}(d|p(c,\theta)) \prod_i C_i(\theta_i, \sigma_i) \,. \tag{2}$$

The nuisance parameters should be removed through their constraints, which depends on the type of underlying uncertainty. Systematic uncertainties typically correspond to auxiliary measurements with large amounts of data, so they are described by Gaussians. They include, for instance, the luminosity or the lepton and photon reconstruction. Theory uncertainties, for instance arising from unknown higher orders or PDFs, are described by flat likelihoods [20]. We can remove the nuisance parameters $\theta$ by profiling,

$$L_{\text{prof}}(\theta) = \max_{\theta} \mathcal{N}(d|p(c,\theta)) \prod_i \mathcal{N}(\theta_i, \sigma_i) \prod_j \mathcal{F}(\theta_j, \sigma_j) \,. \tag{3}$$

The product of Gaussians becomes a single Gaussian with total width $\sigma_{\text{syst}} = \sum_i \sigma_{syst,i}^2$, while the theory uncertainties lead to a shift of the theory prediction by $\sigma_{\text{theo}} = \sum_i \sigma_{theo,i}$ towards the data. By profiling, theory uncertainties are added linearly, while systematics are added quadratically. The final likelihood for signal strengths becomes

$$\sqrt{-2\log L_{\text{prof}}(\theta)} = \begin{cases} (p + \sigma_{\text{theo}} - d)/\sigma_{\text{syst}} & d < p - \sigma_{\text{theo}} \\ 0 & d \in [p - \sigma_{\text{theo}}, p + \sigma_{\text{theo}}] \\ (p - \sigma_{\text{theo}} - d)/\sigma_{\text{syst}} & d > p + \sigma_{\text{theo}} \,. \end{cases} \tag{4}$$

**Higgs-gauge sector**

The Higgs and di-boson data also includes measurements of signal plus background rates [21, 22]. Now, the exclusive likelihood in Eq.(1) cannot be simplified, and we need to profile over a product of Poisson, Gaussian and flat distributions. If we are interested in the observed signal number $s = d - b$, we can express the corresponding Poisson terms as a function of $\tilde{s} = p - b$

$$\text{Poiss}(d|p) = \text{Poiss}(\tilde{s}|d, bk) = \frac{e^{-(\tilde{s}+bk)}(\tilde{s} + bk)^d}{d!} \,. \tag{5}$$

Furthermore, we define a generalized $\chi^2$, which vanishes when data and prediction match,

$$\begin{aligned} \chi^2 &= -2\log \frac{\text{Poiss}(\tilde{s}|d, bk)\text{Poiss}(b_{CR}|bk)}{\text{Poiss}(\tilde{s}|p, bk)\text{Poiss}(bk|bk)} \\ &= -2\left[ (d-p)\log(\tilde{s} + bk) + (b_{CR} - bk)\log(bk) + \log\left( \frac{p!}{d!} \frac{(bk)!}{b_{CR}!} \right) \right] \,. \end{aligned} \tag{6}$$

Next, we profile over the expected background $b$. Performing this maximization for each data point would be inefficient, we approximate this contribution by splitting it into two parts

$$\begin{aligned} \log L_{\text{Poiss},d}(\tilde{s}|d, b_{CR}) &= d - (\tilde{s}_\sigma + b_{CR})\log(\tilde{s}_\sigma + b_{CR}) + \log\frac{(\tilde{s}_\sigma + b_{CR})!}{d!} \\ \log L_{\text{Poiss},b}(\tilde{s}|d, b_{CR}) &= b_{CR} - (d - \tilde{s}_\sigma)\log(d - \tilde{s}_\sigma) + \log\frac{(d - \tilde{s}_\sigma)!}{b_{CR}!} \,. \end{aligned} \tag{7}$$

We incorporate the effect of the flat nuisance parameters by introducing the shifted signal $\tilde{s}_\sigma = \tilde{s} \pm \sigma_{\text{theo}}$, where the sign is chosen such that the signal is shifted towards data. The last missing piece are the Gaussian systematics. These are analogous to the signal strengths and can be computed as

$$-2 \log L_{\text{Gauss}}(\tilde{s}|d, b_{CR}) = \frac{(d - b_{CR} - \tilde{s}_\sigma)^2}{\sum_{\text{syst}} (\sigma_{d,i} - \sigma_{b,i})^2} \,. \tag{8}$$

Finally, we combine all these contributions using the approximate formula

$$\frac{1}{L_{\text{full}}} \approx \frac{1}{L_{\text{Gauss}}} + \frac{1}{L_{\text{Poiss},b}} + \frac{1}{L_{\text{Poiss},d}} \,, \tag{9}$$

which is exact in the fully Gaussian case and has been shown to give excellent results in previous SFITTER analyses [23].

**Correlations**

A vital aspect of the SFITTER likelihood is that all systematic uncertainties of the same type and experiment are fully correlated between measurements. This is done through a correlation matrix with off-diagonal entries

$$C_{ij} = \frac{\sum_{\text{syst}} \rho_{ij} \sigma_{i,\text{syst}} \sigma_{j,\text{syst}}}{\sigma_{i,\text{exp}} \sigma_{j,\text{exp}}} \qquad \text{with} \qquad \sigma_{i,\text{exp}}^2 = \sum_{\text{syst}} \sigma_{i,\text{syst}}^2 + \sum_{\text{Poiss}} \sigma_{i,\text{Poiss}}^2 \,, \tag{10}$$

where the indices $i, j$ run over all measurements, and we choose $\rho_{ij} = 0.99$ to ensure the invertibility of the correlation matrix.

**Fast GPU-evaluation**

Computationally, the most costly part of computing likelihoods is predicting the rate as a function of the Wilson coefficients. Using the quadratic dependence of the rate on the Wilson coefficients we can build suitable matrices to write the rates as a bilinear operation,

$$p_i^{(b)} = W_{ijk} C_j^{(b)} \tilde{C}_k^{(b)} + B_i \tag{11}$$

where $b$ is a batch index, $i$ runs over all observations, and $j$ and $k$ over the Wilson coefficients. $\tilde{C}$ is padded with a 1 to allow for linear dependencies. As this is a very simple tensor operation, it can be accelerated considerably by moving it to a GPU. Similar optimizations can be applied to other parts of the likelihood, like the prediction of branching ratios in the Higgs fit. By performing the computation in PYTORCH, we also get access to its gradients which allows us to use more efficient maximization strategies.

## 2.2   Improved likelihood sampling

Neural importance sampling (NIS) offers a way to sample much more efficiently from the SFITTER likelihood. However, likelihood sampling defines a different set of challenges compared to MadNIS applied for phase space sampling [15,17,18]: the likelihoods can be strongly correlated and multi-modal, and we do not have access to physics-defined mappings and multi-channeling. Instead, we account for these properties during the training of the normalizing flow. Furthermore, sampling alone is not sufficient to get smooth profile-likelihoods, so it has to be combined with a maximization procedure. We divide our novel training and fitting method into five steps:

1. **Pre-scaling**: obtain an approximate sample from the likelihood to estimate component-wise means and standard-deviations;
2. **Pre-training**: use that sample to train a normalizing flow to give the NIS training a better starting point;
3. **Training**: run a NIS training, including annealed importance sampling and buffered training to improve the convergence;
4. **Sampling**: use the trained normalizing flow to generate weighted samples. Make histograms and keep track of the maximal likelihood in each bin;
5. **Maximizing**: use gradients to further improve the estimate of the profile likelihood computed during sampling.

**Pre-scaling**

To efficiently sample from a distribution, we shift and scale the parameter space such that it is centered around zero with unit standard deviation. This helps for MCMC as well as ML sampling. To this end, we draw a batch of samples that approximates our target distribution. As standard MCMC requires a burn-in and can only be parallelized as independent Markov chains, we use annealed importance sampling [24]. It combines the advantages of Markov chains and importance sampling by gradually transforming a tractable base distribution $p_0(x)$ to the target $p_T(x)$ through intermediate log-linear distributions

$$\log p_t(x) = (1 - \beta_t) \log p_0(x) + \beta_t \log p_T(x) \qquad \text{with} \qquad \beta_t = \frac{t}{T} \quad \text{and} \quad t = 1, \dots, T \,, \quad (12)$$

We initialize the sampling by drawing samples from a Gaussian base distribution,

$$x_0 \sim p_0(x_0) = \mathcal{N}_{0,\sigma}(x_0) \qquad \text{and} \qquad w_0 = 1 \,. \qquad (13)$$

Its standard deviation $\sigma$ should not be much narrower than the target distribution, but does not require much tuning otherwise. Only in cases where the width of the distribution differs by orders of magnitude in different directions, it is necessary to coarsely initialize the scaling by hand. Then the following steps are repeated for all $t$

1. transport the samples to the next $t$-distribution through re-weighting with

$$w_t = \frac{p_t(x_{t-1})}{p_{t-1}(x_{t-1})} w_{t-1} \,, \qquad (14)$$

2. evolve the samples according to the distribution $p_t$ using one or more Metropolis-adjusted MCMC steps, $x_t = \text{MCMC}(x_{t-1})$ .

At the end, we arrive at weighted samples $x_T$ with weights

$$w \equiv w_T = \prod_{t=1}^{T} \frac{p_t(x_{t-1})}{p_{t-1}(x_{t-1})} \,. \qquad (15)$$

To get weights close to one, the number of $t$-steps has to be sufficiently large. Because the samples are uncorrelated, the method can be easily parallelized. For our MCMC steps we use a Gaussian proposal distribution with step size $\tau$,

$$q(x'|x) \propto \exp\left[ -\frac{(x'-x)^2}{4\tau} \right] \,. \qquad (16)$$

We could use the available gradient information about our target distribution for more sophisticated methods, like Langevin or Hamiltonian Markov chains, but the additional cost of

evaluating the gradient is not justified by the improvement in sampling. To ensure detailed balance or unbiased sampling from the distribution $p$, we use the Metropolis-Hastings algorithm and accept samples with the probability

$$P_{\text{accept}}(x', x) = \min\left[1, \frac{p(x')\,q(x|x')}{p(x)\,q(x'|x)}\right].$$

(17)

The compromise between acceptance $a$ and parameter exploration is determined by the step size $\tau$. Since we do not use the samples generated during pre-scaling later, we can make the step size adaptive without having to worry about biasing our samples. After every MCMC step and for a given target acceptance $a_{\text{target}}$, we update the step size as

$$\tau \leftarrow \tau \times 2^{\min(r,1)} \quad \text{with} \quad r = \frac{a}{a_{\text{target}}} - 1.$$

(18)

To improve the weights of the generated samples, we can use re-sampling [25] between steps 1 and 2, whenever the effective sample size

$$N_t^{\text{eff}} = \frac{\left(\sum_i w_t^i\right)^2}{\sum_i (w_t^i)^2}$$

(19)

drops below a threshold. In that case we draw new samples from the weighted samples $(w_t^i, x_t^i)$, using the normalized weights as probabilities. After re-sampling, all samples are assigned the same weight,

$$w'^i_t = \frac{1}{N}\sum_{i=1}^{N} w_t^i.$$

(20)

While this can initially lead to some degeneracy, the MCMC steps allow samples to move away from their common starting point. This allows the method to focus more on promising samples and leads to a much narrower weight distribution.

At the end of the pre-scaling, we are left with weighted samples $(w^i, x^i)$. We use these to calculate the component-wise means $\mu_k$ and standard deviations $\sigma_k$, and from now on use the standardized space coordinates $(x_k^i - \mu_k)/\sigma_k$.

**Flow pre-training**

In MadNIS we combine neural importance sampling with pre-defined phase-space mappings, which incorporate our physics knowledge. To sample over Wilson coefficients we do not have such mappings.

A standard NIS training, where we throw samples into the parameter space and then optimize the network using these samples, will now fail or converge very slowly. Instead, we use a small set of samples from our target distribution to pre-train the flow. For this purpose we can use the samples from the pre-scaling. We perform a final re-sampling on them and evolve them for a number of MCMC steps to get unweighted training data. This way we avoid working with a too small pre-training sample. The flow is then trained on these samples using a standard log-likelihood loss,

$$\mathcal{L} = -\log g_\theta(x),$$

(21)

where $g_\theta(x)$ is the tractable probability distribution encoded by the normalizing flow with trainable parameters $\theta$. Because we typically perform the pre-training on a relatively small

datasets, like 10 batches of 1024 samples, we evolve the samples using further MCMC steps after every batch.

While a short pre-training done this way is not sufficient to learn the target distribution with high precision, it is sufficient to give the main NIS training a sufficiently good starting point.

**Flow training**

Annealed importance sampling can also be used to make the training of flow networks for neural importance sampling more efficient and avoid common problems like mode collapse or inefficient training for distributions with narrow features [26]. Consider a variance loss

$$\mathcal{L} = \left\langle \frac{p(x)^2}{g_\theta(x)q(x)} \right\rangle_{x \sim q(x)} \tag{22}$$

with a proposal distribution $q(x)$. In a regular NIS training we choose $q(x) = g_\theta(x)$, such that we can use our flow to generate the training samples online. However, this proposal does not minimize the variance of the loss. i.e. the variance of the variance. The optimal proposal distribution is

$$q(x) = f_\theta(x) \equiv \frac{p(x)^2}{g_\theta(x)} , \tag{23}$$

such that the expectation value in Eq.(22) becomes trivial. We can get weighted samples from this distribution using annealed importance sampling, using the flow $g_\theta(x)$ as the base distribution and $f_\theta(x)$ as the target distribution. After setting $q(x) = f_\theta(x)$ in Eq.(22), we can rewrite the gradients as

$$\nabla_\theta \mathcal{L} = \left\langle \frac{\nabla_\theta f_\theta(x)}{f_\theta(x)} \right\rangle_{x \sim q(x)} = \left\langle \nabla_\theta \log f_\theta(x) \right\rangle_{x \sim q(x)} = -\left\langle \nabla_\theta \log g_\theta(x) \right\rangle_{x \sim q(x)} . \tag{24}$$

After annealed importance sampling and in terms of its weights $w$, we can write the loss as

$$\mathcal{L}_{\text{online}} = -\left\langle w \log g_\theta(x) \right\rangle_{x \sim g_\theta(x)} + \text{const} . \tag{25}$$

To improve the stability of this loss evaluation, we apply a batch-wise normalization of the weights and limit the effect of very large weights using a modified weight function, which is approximately linear with unit slope for small weights and logarithmic for $\gtrsim 10$,

$$w' = \alpha \log \left( \frac{w}{\alpha} + 1 \right) \quad \text{with} \quad \alpha = 30 . \tag{26}$$

As in MadNIS, we speed up the training using a buffered sampling step. However, instead of training the network on weighted samples taken from the buffer with uniform probability, we instead follow the method proposed in Ref. [26]. We draw samples from the buffer with a probability proportional to their importance sampling weight, such that the weight update is performed on approximately unweighted events. We only have to account for the change in the network parameters at the time of sampling $\theta'$ and at the time of optimization $\theta$. The loss function of a buffered training step then reads

$$\mathcal{L}_{\text{buffered}} = \left\langle \frac{q_\theta(x)}{q_{\theta'}(x)} \log q_\theta(x) \right\rangle . \tag{27}$$

After each buffered sampling step, we update the buffered weights using the $q$-ratio to reduce the bias of the buffered samples compared to the target distribution.

**Sampling**

After the network is trained, we can use it to draw weighted samples from our target distribution. We are interested in 1D and 2D marginal distributions which we generate as histograms. Furthermore, we are interested in the 1D and 2D profile likelihoods, i.e. the maximum of the likelihood when one or two model parameters are fixed.

We fix the binning of the histograms for marginalization and profiling after the first batch of generated samples. We then generate 1D histograms for all parameters and 2D histograms for all parameter pairs. For the profile likelihood, we select the sample with the highest likelihood in each bin for each batch of samples. We then change the one or two fixed parameters such that the point is in the center of the bin and re-evaluate the likelihood for the updated parameter point. We then choose a fixed number of points with the highest likelihoods for each bin from all batches.

**Maximizing**

The sampling provides us with points close to the maximum likelihood, with one or two parameters fixed. To further improve the profile likelihood and to reduce the effects from limited statistics, we use these points as the starting point for a maximization based on gradient-ascent through automatic differentiation in PyTorch.

In simple cases it is sufficient to use fast gradient-ascent based optimizers like ADAM. For more complex likelihoods we use the L-BFGS optimizer, a second-order optimization method, as it allows for more precise estimation of the maximum.

## 3 SMEFT

The SMEFT [27–30] provides a perfect framework for model agnostic searches for new physics arising from particles too heavy to be produced directly. We parametrize these new physics contributions via Wilson coefficients $C_k$ and higher-dimensional operators $\mathcal{O}_k$. They are included in the SMEFT Lagrangian by expanding it in inverse powers of the new physics scale $\Lambda$, respecting the symmetries and field content of the SM

$$\mathcal{L}_{\text{SMEFT}} = \mathcal{L}_{\text{SM}} + \sum_k \frac{C_k}{\Lambda^2} \mathcal{O}_k \ . \tag{28}$$

In our analysis we ignore lepton-number violating operators, which removes any dimension-5 operators, and CP-violating operators. To search for those we prefer dedicated, optimal analyses [31–33]. Moreover, we truncate our SMEFT expansion at dimension six, motivated by the assumption that the suppression in terms of $\Lambda$ translates from the Lagrangian to the physical observables by integrating out heavy particle contributions. For limitations induced by this truncations we refer for instance to Refs. [34, 35].

The number of operators can be further reduced by restricting the analysis to specific classes of processes. For our analysis we include top quark observables, as well as Higgs, di-boson and electroweak precision observables (EWPOs). SMEFT operator contributions are computed up to quadratic order for top, Higgs and di-boson observables, while only linear terms are included for EWPOs. Additional assumptions, such as those regarding flavor structure, depend on the type of data considered.

| Operator | Definition | Operator | Definition |
|---|---|---|---|
| $\mathcal{O}_{Qq}^{1,8}$ | $(\bar{Q}\gamma_\mu T^A Q)\,(\bar{q}_i \gamma^\mu T^A q_i)$ | $\mathcal{O}_{tu}^{8}$ | $(\bar{t}\gamma_\mu T^A t)\,(\bar{u}_i \gamma^\mu T^A u_i)$ |
| $\mathcal{O}_{Qq}^{1,1}$ | $(\bar{Q}\gamma_\mu Q)\,(\bar{q}_i \gamma^\mu q_i)$ | $\mathcal{O}_{tu}^{1}$ | $(\bar{t}\gamma_\mu t)\,(\bar{u}_i \gamma^\mu u_i)$ |
| $\mathcal{O}_{Qq}^{3,8}$ | $(\bar{Q}\gamma_\mu T^A \tau^I Q)\,(\bar{q}_i \gamma^\mu T^A \tau^I q_i)$ | $\mathcal{O}_{td}^{8}$ | $(\bar{t}\gamma^\mu T^A t)\,(\bar{d}_i \gamma_\mu T^A d_i)$ |
| $\mathcal{O}_{Qq}^{3,1}$ | $(\bar{Q}\gamma_\mu \tau^I Q)\,(\bar{q}_i \gamma^\mu \tau^I q_i)$ | $\mathcal{O}_{td}^{1}$ | $(\bar{t}\gamma^\mu t)\,(\bar{d}_i \gamma_\mu d_i)$ |
| $\mathcal{O}_{Qu}^{8}$ | $(\bar{Q}\gamma^\mu T^A Q)\,(\bar{u}_i \gamma_\mu T^A u_i)$ | $\mathcal{O}_{Qd}^{1}$ | $(\bar{Q}\gamma^\mu Q)\,(\bar{d}_i \gamma_\mu d_i)$ |
| $\mathcal{O}_{Qu}^{1}$ | $(\bar{Q}\gamma^\mu Q)\,(\bar{u}_i \gamma_\mu u_i)$ | $\mathcal{O}_{tq}^{8}$ | $(\bar{q}_i \gamma^\mu T^A q_i)\,(\bar{t}\gamma_\mu T^A t)$ |
| $\mathcal{O}_{Qd}^{8}$ | $(\bar{Q}\gamma^\mu T^A Q)\,(\bar{d}_i \gamma_\mu T^A d_i)$ | $\mathcal{O}_{tq}^{1}$ | $(\bar{q}_i \gamma^\mu q_i)\,(\bar{t}\gamma_\mu t)$ |
| $\mathcal{O}_{\phi Q}^{1}$ | $(\phi^\dagger i \overleftrightarrow{D}_\mu \phi)\,(\bar{Q}\gamma^\mu Q)$ | $^\ddagger\mathcal{O}_{tB}$ | $(\bar{Q}\sigma^{\mu\nu} t)\,\widetilde{\phi}\, B_{\mu\nu}$ |
| $\mathcal{O}_{\phi Q}^{3}$ | $(\phi^\dagger i \overleftrightarrow{D}_\mu^I \phi)\,(\bar{Q}\gamma^\mu \tau^I Q)$ | $^\ddagger\mathcal{O}_{tW}$ | $(\bar{Q}\sigma^{\mu\nu} t)\,\tau^I \widetilde{\phi}\, W_{\mu\nu}^I$ |
| $\mathcal{O}_{\phi t}$ | $(\phi^\dagger i \overleftrightarrow{D}_\mu \phi)\,(\bar{t}\gamma^\mu t)$ | $^\ddagger\mathcal{O}_{bW}$ | $(\bar{Q}\sigma^{\mu\nu} b)\,\tau^I \phi\, W_{\mu\nu}^I$ |
| $^\ddagger\mathcal{O}_{\phi tb}$ | $(\widetilde{\phi}^\dagger i D_\mu \phi)\,(\bar{t}\gamma^\mu b)$ | $^\ddagger\mathcal{O}_{tG}$ | $(\bar{Q}\sigma^{\mu\nu} T^A t)\,\widetilde{\phi}\, G_{\mu\nu}^A$ |

Table 1: List of the 22 independent operators contributing to our top observables. They are related to the Warsaw basis in the Appendix of Ref. [7].

### 3.1 Top sector

The operator basis for the top sector analysis follows Ref. [7], with non-hermitian operators denoted as $^\ddagger\mathcal{O}$. We impose a $U(2)$ flavor symmetry on the first and second quark generations, as most top observables are blind to the flavor of light quarks,

$$q_i = (u_L^i, d_L^i) \qquad u_i = u_R^i, d_i = d_R^i \quad \text{for} \quad i = 1,2$$
$$Q = (t_L, b_L) \qquad t = t_R, b = b_R\,. \tag{29}$$

All quark masses except for the top mass are taken to be zero, which leaves 22 independent operators listed in Tab 1.

The top-sector operators are divided into three distinct categories. The top row describes four-fermion currents with $RR$ and $LL$ helicity structure, while the center row operators include a $RL$ and $LR$ helicity flips. These are mainly constrained by observables involving top pairs, such as $t\bar{t}$ and associated $t\bar{t}W, t\bar{t}Z$ production. The final set of operators couple heavy quarks to gauge bosons. For convenience, we make use of additional relations arising from gauge invariance

$$C_{\phi Q}^- = C_{\phi Q}^1 - C_{\phi Q}^3 \qquad \text{and} \qquad C_{tZ} = c_w C_{tW} - s_w C_{tB}. \tag{30}$$

With these definitions we choose $C_{\phi Q}^-, C_{\phi Q}^3, C_{tW}$ and $C_{tZ}$ as our degrees of freedom. To fully constrain them, we include single top production, associated $tZ$ and $tW$ production, as well as top decay observables. A detailed description of the observables included in our dataset and the impact of the different Wilson coefficients is given in Ref. [12]. As part of this study we also showed that we can use published likelihoods by ATLAS and CMS, including detailed uncertainty information, in our SFITTER analysis.

### 3.2 Higgs-gauge sector

For the analysis of Higgs, di-boson and EWPOs data we adopt the HISZ operator basis based on physics arguments from Run1 and Run2 [21,22,36]. All operators are listed in Tab 2. They are, again, split into several categories. The top row lists all operators affecting the Higgs

| Operator | Definition | Operator | Definition |
|---|---|---|---|
| $\mathcal{O}_{GG}$ | $\phi^\dagger \phi\, G^a_{\mu\nu} G^{a\mu\nu}$ | $\mathcal{O}_{WW}$ | $\phi^\dagger\, \hat{W}_{\mu\nu}\hat{W}^{\mu\nu}\,\phi$ |
| $\mathcal{O}_{BB}$ | $\phi^\dagger\, \hat{B}_{\mu\nu}\hat{B}^{\mu\nu}\,\phi$ | $\mathcal{O}_W$ | $(D_\mu\phi)^\dagger \hat{W}^{\mu\nu}(D_\nu\phi)$ |
| $\mathcal{O}_B$ | $(D_\mu\phi)^\dagger \hat{B}^{\mu\nu}(D_\nu\phi)$ | $\mathcal{O}_{BW}$ | $\phi^\dagger \hat{B}_{\mu\nu}\hat{W}^{\mu\nu}\,\phi$ |
| $\mathcal{O}_{\phi 1}$ | $(D_\mu\phi)^\dagger\, \phi\phi^\dagger\, (D^\mu\phi)$ | $\mathcal{O}_{\phi 2}$ | $\frac{1}{2}\partial^\mu(\phi^\dagger\phi)\partial_\mu(\phi^\dagger\phi)$ |
| $\mathcal{O}_{3W}$ | $\mathrm{Tr}\left(\hat{W}_{\mu\nu}\hat{W}^{\nu\rho}\hat{W}^\mu_\rho\right)$ | | |
| $\mathcal{O}^{(1)}_{\phi u}$ | $\phi^\dagger(i\overleftrightarrow{D}_\mu\phi)(\bar{u}_R\gamma^\mu u_R)$ | $\mathcal{O}^{(1)}_{\phi Q}$ | $\phi^\dagger(i\overleftrightarrow{D}_\mu\phi)(\bar{Q}\gamma^\mu Q)$ |
| $\mathcal{O}^{(1)}_{\phi d}$ | $\phi^\dagger(i\overleftrightarrow{D}_\mu\phi)(\bar{d}_R\gamma^\mu d_R)$ | $\mathcal{O}^{(3)}_{\phi Q}$ | $\phi^\dagger(i\overleftrightarrow{D^a_\mu}\phi)\left(\bar{Q}\gamma^\mu \frac{\sigma_a}{2}Q\right)$ |
| $\mathcal{O}^{(1)}_{\phi e}$ | $\phi^\dagger(i\overleftrightarrow{D}_\mu\phi)(\bar{e}_R\gamma^\mu e_R)$ | | |
| $\mathcal{O}_{e\phi,22}$ | $\phi^\dagger\phi\, \bar{L}_2\phi e_{R,2}$ | $\mathcal{O}_{e\phi,33}$ | $\phi^\dagger\phi\, \bar{L}_3\phi e_{R,3}$ |
| $\mathcal{O}_{u\phi,33}$ | $\phi^\dagger\phi\, \bar{Q}_3\tilde{\phi}u_{R,3}$ | $\mathcal{O}_{d\phi,33}$ | $\phi^\dagger\phi\, \bar{Q}_3\phi d_{R,3}$ |
| $\mathcal{O}_{4L}$ | $(\bar{L}_1\gamma_\mu L_2)(\bar{L}_2\gamma^\mu L_1)$ | | |

Table 2: List of the 19 independent operators contributing to our Higgs, di-boson and electroweak observables.

interactions with gauge bosons, all of which can be constrained using Higgs processes, except for $\mathcal{O}_{3W}$, which is constrained by di-boson production and electroweak precision measurements. The next two rows list single-current operators affecting both gauge and Higgs-gauge couplings. These are split into two parts, where $\mathcal{O}^1_{\phi u}, \mathcal{O}^1_{\phi d}, \mathcal{O}^1_{\phi e}, \mathcal{O}^1_{\phi q}, \mathcal{O}^3_{\phi q}$ are flavor universal, while we allow for minimal flavor violation coming from $\mathcal{O}_{e\phi,22}, \mathcal{O}_{e\phi,33}, \mathcal{O}_{u\phi,33}, \mathcal{O}_{d\phi,33}$. Finally, we include the four-lepton operator $\mathcal{O}_{4L}$ which induces a shift in the Fermi constant. As in the top analysis we use an orthogonal combination as our actual degrees of freedom

$$\mathcal{O}_\pm = \frac{\mathcal{O}_{WW} \pm \mathcal{O}_{BB}}{2} \qquad \Rightarrow \qquad f_\pm = f_{WW} \pm f_{BB}\,. \tag{31}$$

This way only $\mathcal{O}_+$ contributes to the $H\gamma\gamma$ interaction.

### 3.3 Combined analysis

It is not possible to naively combine our two top and Higgs-gauge operator bases into a combined analysis, because they are based on different flavor assumptions. First, we need to convert the HISZ-operators in Tab. 2 into the Warsaw basis using the expressions derived in the Appendix of Ref. [37]. In the Warsaw basis, illustrated in Tab. 3, we can align the flavor structures of the two sectors. After that, the combined analysis including comprehensive and fully correlated uncertainties is just a numerical problem.

On the physics side, the combination of the two sectors defines a few distinct bridges between the largely de-correlated sets of operators and observables. On the observable side, the obvious bridges are associated top (pair) production with Higgs and gauge bosons. For SFITTER, the historic split is to include $t\bar{t}H$ production in the Higgs-gauge sector, because it cannot be separated from gluon-fusion Higgs production. On the other hand, for instance $t\bar{t}Z$ production is part of the top sector analysis, leading to an artificial split motivating a combination of the two sectors. In addition to the top Yukawa operator, the main bridge between the two sectors is $\mathcal{O}_{tG}$ [10], originally part of the SFITTER top sector analysis.

While the combination of the top and Higgs-gauge sectors to a proper global SMEFT analysis is a clear physics goal, it leads to technical complications. The requirements on the treatment of the likelihoods in the two sectors are very different. For the Higgs-gauge sector kinematic distributions, for instance in di-boson and $VH$ production, have shown to be extremely

| Operator | Definition | Operator | Definition |
|---|---|---|---|
| $\mathcal{O}_{\phi G}$ | $\phi^\dagger \phi G_{\mu\nu}^A G^{A\mu\nu}$ | $\mathcal{O}_W$ | $\varepsilon^{IJK} W_\mu^{I\nu} W_\nu^{J\rho} W_\rho^{K\mu}$ |
| $\mathcal{O}_{\phi B}$ | $\phi^\dagger \phi B_{\mu\nu} B^{\mu\nu}$ | $\mathcal{O}_{\phi W}$ | $\phi^\dagger \phi W_{\mu\nu}^I W^{I\mu\nu}$ |
| $\mathcal{O}_{\phi WB}$ | $\phi^\dagger \tau^I \phi W_{\mu\nu}^I B^{\mu\nu}$ | | |
| $\mathcal{O}_{\phi\square}$ | $(\phi^\dagger \phi)\square(\phi^\dagger \phi)$ | $\mathcal{O}_{\phi D}$ | $(\phi^\dagger D^\mu \phi)^*(\phi^\dagger D^\mu \phi)$ |
| $\mathcal{O}_{\phi e}$ | $(\phi^\dagger i\overleftrightarrow{D}_\mu \phi)(\bar{e}_i \gamma^\mu e_i)$ | $\mathcal{O}_{\phi b}$ | $(\phi^\dagger i\overleftrightarrow{D}_\mu \phi)(\bar{b}_i \tau^I \gamma^\mu b_i)$ |
| $\mathcal{O}_{\phi d}$ | $\sum_{i=1}^{2}(\phi^\dagger i\overleftrightarrow{D}_\mu \phi)(\bar{d}_i \gamma^\mu d_i)$ | $\mathcal{O}_{\phi u}$ | $\sum_{i=1}^{2}(\phi^\dagger i\overleftrightarrow{D}_\mu \phi)(\bar{u}_i \gamma^\mu u_i)$ |
| $\mathcal{O}_{\phi q}^{(1)}$ | $\sum_{i=1}^{2}(\phi^\dagger i\overleftrightarrow{D}_\mu \phi)(\bar{q}_i \gamma^\mu q_i)$ | $\mathcal{O}_{\phi q}^{(3)}$ | $\sum_{i=1}^{2}(\phi^\dagger i\overleftrightarrow{D}_\mu \phi)(\bar{q}_i \tau^I \gamma^\mu q_i)$ |
| $\mathcal{O}_{\phi l}^{(1)}$ | $(\phi^\dagger i\overleftrightarrow{D}_\mu \phi)(\bar{l}\gamma^\mu l)$ | $\mathcal{O}_{\phi l}^{(3)}$ | $(\phi^\dagger i\overleftrightarrow{D}_\mu^I \phi)(\bar{l}\tau^I \gamma^\mu l)$ |
| $\mathcal{O}_{d\phi,33}$ | $(\phi^\dagger \phi)(\bar{Q}_3 b\phi)$ | $\mathcal{O}_{u\phi,33}$ | $(\phi^\dagger \phi)(\bar{Q}_3 t\phi)$ |
| $\mathcal{O}_{e\phi,22}$ | $(\phi^\dagger \phi)(\bar{l}_2 \mu\phi)$ | $\mathcal{O}_{e\phi,33}$ | $(\phi^\dagger \phi)(\bar{l}_3 \tau\phi)$ |
| $\mathcal{O}_{ll}$ | $(\bar{l}\gamma_\mu l)(\bar{l}\gamma^\mu l)$ | | |

Table 3: Operators in the Warsaw basis for Higgs, di-boson and electroweak sectors. These 21 degrees of freedom and those from the Top sector are included in the combined global analysis.

powerful and largely statistics-limited. Theory and systematic uncertainties and their correlations are less important in kinematic tails, but under-fluctuations in low-statistics regimes have to be accommodated, leading to significant differences between profiled and marginalized single-operator limits [36]. In contrast, the top sector analysis is driven by (associated) top pair production, a QCD process with large rates. Even kinematic tails are heavily populated, and the correlated theory uncertainties are key to an otherwise mostly Gaussian analysis. Again, the combination of a flat theory uncertainty with a Gaussian nuisance parameter leads to sizeable differences between profiled and marginalized results. All of these differences pose technical requirements on the combined analysis, where the subtleties of both sectors have to be treated correctly.

# 4 Results

The focus of this paper is not a new SMEFT analysis, but a new combination of physics assumptions and ML-methods to evaluate the fully exclusive likelihood and extract marginalized and profiled likelihoods. We start by illustrating this technique for a toy example in Sec. 4.1 and then show the improved global SFITTER analysis in Sec. 4.2.

## 4.1 Toy example

As a first toy model we define two two-dimensional concentric spirals as

$$\begin{pmatrix} r \\ \phi \end{pmatrix} = \pm \begin{pmatrix} t \\ 2\pi t \end{pmatrix} \quad \text{with} \quad t \in [0.36, 1.93]. \tag{32}$$

We turn this spiral into a probability distribution by imposing a uniform distribution along the length of the spiral, and smearing with a Gaussian distribution with a constant width $\sigma$. We then perform training, marginalization and profiling for this distribution with the hyperparameters given in the Appendix. The results for two Gaussian widths are shown in Fig. 1.

Sampling and profiling work well in both cases, and the ML-sampler has no problems resolving the complex structure of the spiral. The gradients in the maximization pass give us a completely smooth profiled likelihood.

For the narrow width, we see a secondary peak in the weight distribution for small weights. This is caused by the network interpolating into the low-probability regions of the distribution in places where it is not able to learn the distribution perfectly. We confirm this by plotting the distribution of samples with $w < 0.1$ in the bottom left panel of Fig. 1.

As a second toy model consider a 2-dimensional Gaussian mixture with a large central peak and four smaller peaks in the corners,

$$p(x) \propto 10 \times \mathcal{N}(x; \mu = (0,0), \sigma = 1) + \sum_{\mu_{1,2}=\pm d} \mathcal{N}(x; \mu = (\mu_1, \mu_2), \sigma = 0.5). \qquad (33)$$

We start with the more compact spacing $d = 4$ and show the results in the upper panels of Fig. 2. Even when we set the distribution of the initial samples during pre-scaling to the width of the central peak, we find that the ML-sampling maps out all four outer peaks.

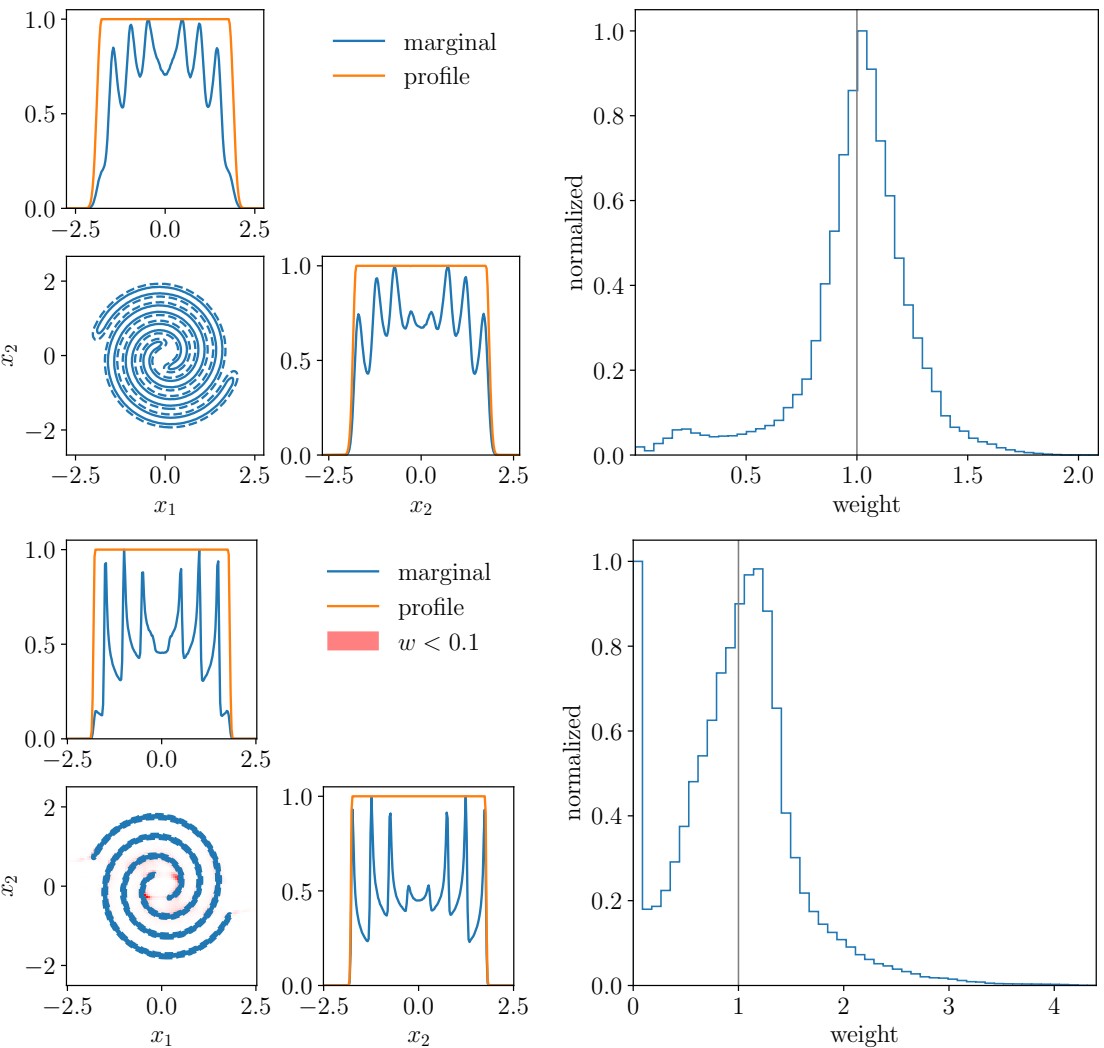

Figure 1: Profiled and marginalized likelihoods, 68% and 95% confidence regions, and weight distributions for the spiral toy distribution with widths $\sigma = 0.007$ (upper) and $\sigma = 0.0005$ (lower).

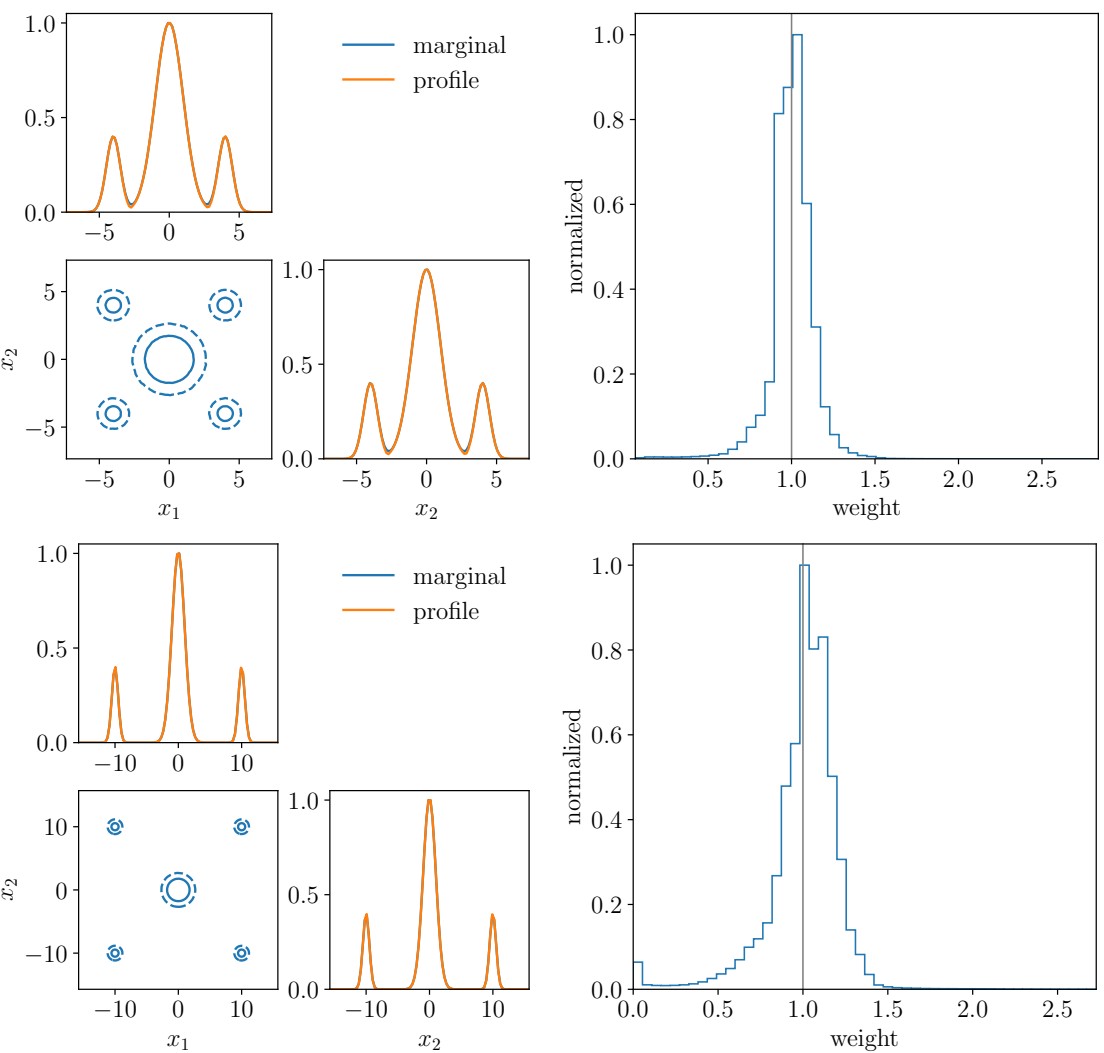

Figure 2: Profiled and marginalized likelihoods, 68% and 95% confidence regions, and weight distributions for the Gaussian mixture model with $d = 4$ (upper) and $d = 10$ (lower).

This gets more difficult when we increase the distance from the center to the outer peaks. Around $d = 8$, we start to observe mode collapse, so some of the outer peaks are no longer found. However, this can be easily prevented by increasing the spread of the initial samples during pre-scaling. With that adjustment, the ML-sampling easily learn the distribution for $d = 10$ without either very large or very small sample weights, as demonstrated in the lower panels of Fig. 2. We even find that the ML-sampling still learns the distribution if we make all peaks narrower by a factor of ten, albeit at the cost of more samples with very small weights.

## 4.2  SFITTER likelihood

Moving on to the LHC, we begin by studying the individual sectors separately to prove the validity of our method, while simultaneously presenting the improvements we gain from having access to the gradients of our likelihood. Finally, we display the full strength of our methods by applying them to the much higher dimensional combined analysis. In particular, we focus on the improvements on the required computational time compared to previous SFITTER analyses.

**Top sector**

We begin with the top sector, since its likelihood should be particularly simple and as such provides the perfect environment to study some basic features and technical improvements. For reference, all correlations for the 22 Wilson coefficients can be found in Fig. 6 in the Appendix.

In Fig. 3 we show correlations for a few selects pairs of Wilson coefficients which display interesting patterns. In the upper left panel we show the correlations between $C_{tW}$ and $C_{bW}$, with a slight correlation and the typical flat profile likelihoods induced by theory uncertainties. While the 1-dimensional profiled and marginalized likelihoods have a similar width, the marginalization leads to a well-defined maximum and round edges by construction. The upper right panels illustrates a non-trivial correlation between $C_{\phi Q}^-$ and $C_{\phi t}$, perfectly reproducing earlier SFITTER results. Because of the form of the correlation, the 1-dimensional limits on $C_{\phi Q}^-$ after marginalization are much stronger than after profiling. This reflects a sizeable impact of the implicit bias by integrating over the Wilson coefficients linearly. The bottom left panel shows the correlations between $C_{tG}$ and $C_{tq}^8$ where we see a secondary, but non-degenerate

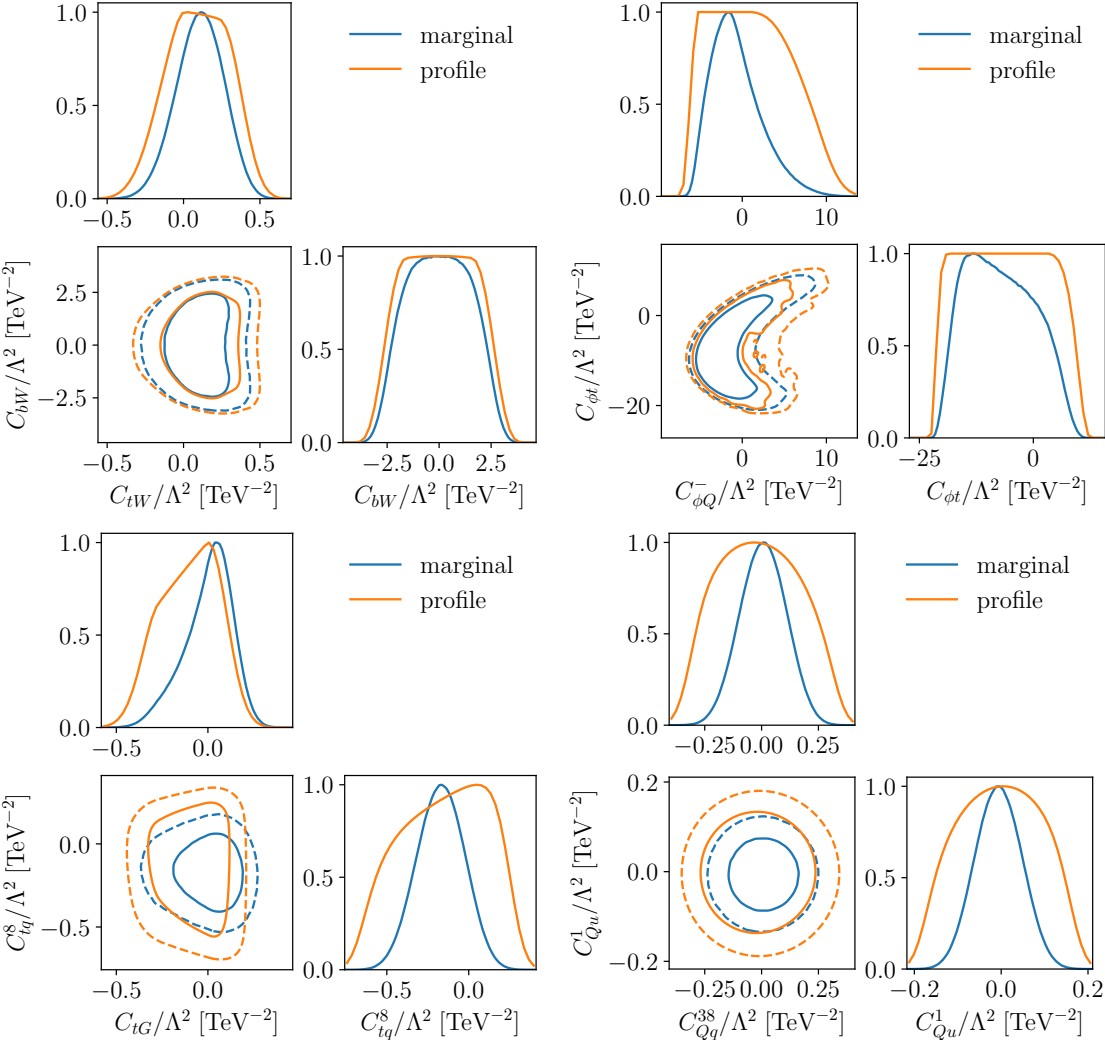

Figure 3: Correlations between select Wilson coefficients in the Top sector, showing results after both profiling (orange) and marginalizing (blue) over the remaining Wilson coefficients.

|                      | Top          | Higgs-gauge | Combined   |
|----------------------|--------------|-------------|------------|
| Dimensions           | 22           | 20          | 42         |
| Training batches     | 100          | 2000        | 6000       |
| Samples              | 10M          | 200M        | 100M       |
| Effective sample size | 7.1M        | 97M         | 21M        |
| Pre-scaling time     | 7s           | 3.5min      | 5.3min     |
| Pre-training time    | 18s          | 1.7min      | 2.5min     |
| Training time        | 36s          | 17.3min     | 1.2h       |
| Sampling time        | 26s          | 14.8min     | 17.6min    |
| Profiling time       | 17.7min      | 24.8min     | 3.7h       |
| Number of CPUs       | 20           | 80          | 120        |
| Accepted samples     | 37M          | 26.4M       | 60M        |
| CPU sampling time    | 29min 49s    | 3h 23min    | 20h 50min  |
| CPU profiling time   | 4min 43s     | 8min 24s    | N/A        |

Table 4: Training and sampling time for the five steps to happiness on an H100 GPU. The CPU times refer to the time needed on the given number of CPUs for the original analyses [12, 36] on the NEMO Cluster. (CPU profiling time for a single 2D correlation plot on a single CPU.)

likelihood structure leading to the enhanced shoulder in both 1-dimensional profile likelihood. The marginalization washes out this effect. Finally, we show the correlation between the two four-fermion operators $C_{Qq}^{38}$ and $C_{Qu}^{1}$. Here, we clearly see the effect of the large theory uncertainties on the top rate measurements, providing much tighter constraints after marginalizing, even though the flat behavior of the profile likelihood is washed out by the additional, correlated directions already profiled away.

The time required to perform the full analysis can be found in the left column of Tab. 4, where also the time needed to perform a traditional SFITTER top analysis is listed. Due to the simplicity of the likelihood, both methods generate many samples efficiently. The entire analysis, including the profiling, runs in minutes on the GPU, and the main difference to CPU-based SFITTER studies is that now the profiling and marginalization lead to equally smooth distributions. Within SFITTER we never achieved this equivalent smoothness, even using 20 CPUs in parallel.

**Higgs-gauge sector**

In the Higgs-gauge sector, the likelihood evaluation becomes harder, because of the more complex and highly correlated shape of the likelihood. This leads to numerical challenges, even though the dimensionality is similar to that of the top sector. Fig. 4 shows two example combinations of Wilson coefficients, while all other correlations can again be found within in Fig. 7 of the Appendix. At this point we use the operator set in the HISZ basis to allow for an easier comparison with previous SFITTER results.

Both plots illustrate the more complicated shape of our likelihood, with two modes now appearing for various Wilson coefficients. In the left panel we see the correlations between $f_+$ and $f_{\phi Q}^{(3)}$ where we see two distinct modes for $f_+$ after profiling and after marginalization. Clearly, one mode is incompatible with small deviations from the SM, and in previous SFITTER analyses the final global analysis was restricted to the SM-like mode. Similarly, all secondary modes arising from sign flips in the Yukawa couplings are removed in the final constraints, since these would require large new physics effects one would likely have observed already.

In the right panels we show the correlation between $f_B$ and $f_{\phi u}^{(1)}$ where after marginaliza-

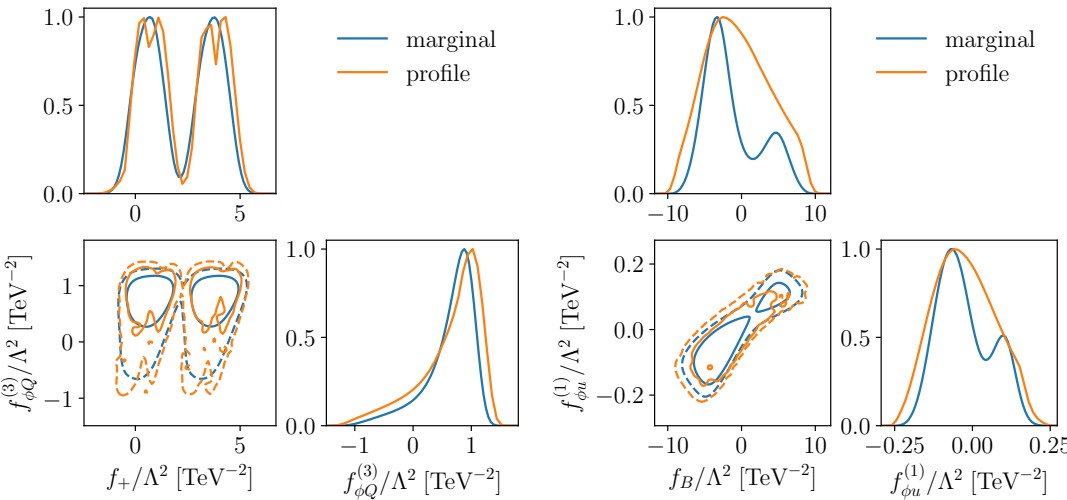

Figure 4: Correlations between two sets of Wilson coefficients in the Higgs sector after profiling (orange) and marginalization (blue).

tion we again find multiple modes for the 1D distributions of our Wilson coefficients while the profiled results do not show this structure. We observe a dip at the SM for both of these coefficients, the cause of which was examined and discussed in great detail in Ref. [36]. To clarify briefly, these differences are the result of an underfluctuation in a kinematic distribution which needs to be explained in the SMEFT. During profiling we always select the most likely point in parameter space, this is typically close to the SM and independent of the dimensionality of the WC parameter space. When marginalizing, however, volume effects appear from the larger space of Wilson coefficients, creating the two peaks at non-SM values.

The timing numbers in the central column of Tab. 4 reflect the numerical challenges from the more complicated correlated Higgs-gauge likelihood. The effective sample size is much larger than for the top sector, and the sampling time for the GPU implementation is also much longer. A full SFITTER analysis needs a total of 80 CPUs for a few hours, and the shape of the profile likelihood of the published results were often quite rugged. Comparatively, we find that even for the more complicated Higgs likelihood our new results are much smoother.

**Combined analysis**

The final step in our analysis is to combine both datasets and perform a full global analysis of all 42 Wilson coefficients. The detailed results in the Warsaw basis can be found in Fig. 8. In Fig. 5 we summarize the results of all these coefficients, split up between the coefficients of the Top sector at the top and those for the Higgs-gauge sector at the bottom. The 1-dimensional profiled and marginalized likelihoods are given in Fig. 9 of the Appendix. In addition to the current limits from Run 2, we also show hypothetical limits assuming that we could reduce the theory uncertainty by a factor of two, leaving the central predictions the same. Because tails of kinematic distributions are statistics-limited, and because there is no tension in the global analysis, the impact of such an improvement is limited to a few Wilson coefficients, like $C_{\phi Q}^1$ or $C_{\phi B}$.

As before, we look at the training and sampling times listed in Tab. 4. While the increased dimensionality of the likelihood requires serious training, the sampling is faster by many orders of magnitude compared to our previous SFITTER studies on 120 CPUs in parallel. The relative speed improvement for sampling on a single GPU is a factor 68 for 120 CPUs vs a single

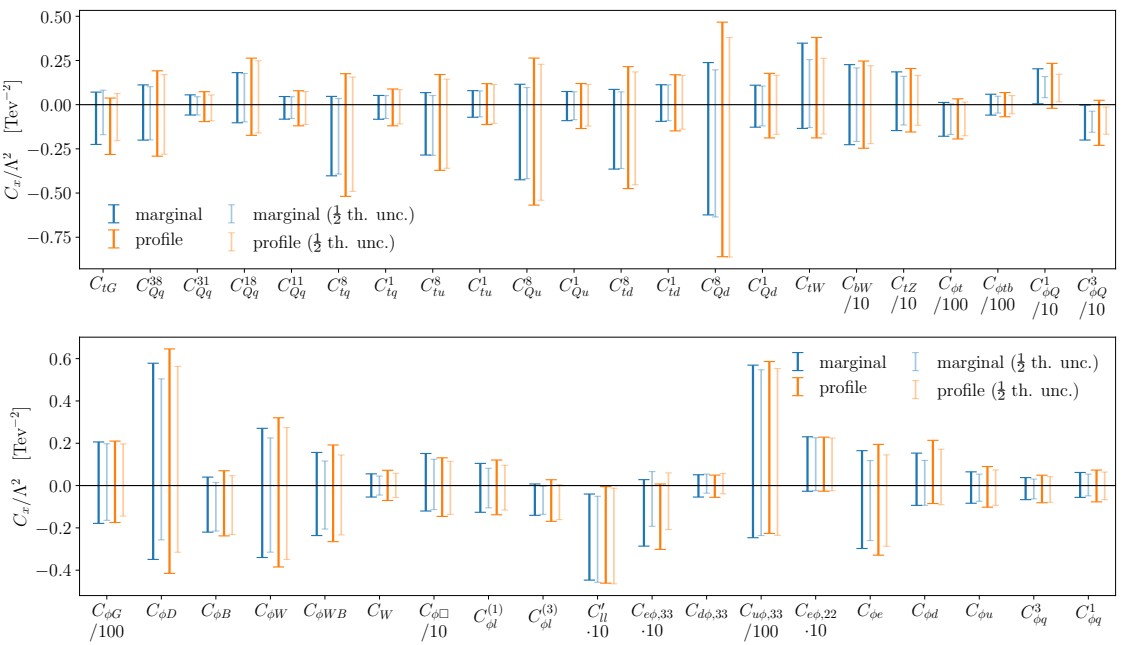

Figure 5: Constraints from the combined SMEFT analysis in the top (top) and Higgs-gauge (bottom) sectors, showing 95% CLs for all 42 Wilson coefficients for both profiled and marginalized likelihoods.

GPU and 8200 for a single CPU vs a single GPU. A complete set of profile likelihoods were, essentially, out of numerical reach for the current CPU implementation.

## 5   Outlook

Global SMEFT analyses are a powerful method telling us to what level the LHC results as a whole agree with the Standard Model (or not). They cover rate measurements and kinematic distributions, defining a large number of measurements with many experimental and theory uncertainties, some of them correlated. Once we combine, for instance, the Higgs-gauge sector with the top sector, the number of Wilson coefficients becomes sizeable as well. This means that the numerical extraction and the analysis of the fully exclusive likelihood is a numerical challenge.

We have shown that applying ML-techniques to learn and evaluate the likelihood allows us to run a global SFITTER analysis on a single GPU in a few hours, rather than on a CPU cluster for a day or more. Our five steps to happiness are inspired by neural importance sampling and MadNIS [15, 17, 18]. They include (i) pre-scaling of the likelihood parameters; (ii) pre-training using a normalizing flow, (iii) training using annealed importance sampling and buffered training, (iv) sampling from the normalizing likelihood flow; and (v) using gradients for efficient profiling. Especially the last step does not only speed up the likelihood evaluation, it also increases its numerical resolution significantly. For future SFITTER analyses, built on a comprehensive and correlated uncertainty treatment, these numerical improvements mean transformative progress.

## Acknowledgements

This research is supported through the KISS consortium (05D2022) funded by the German Federal Ministry of Education and Research BMBF in the ErUM-Data action plan, by the Deutsche Forschungsgemeinschaft (DFG, German Research Foundation) under grant 396021762 – TRR 257: *Particle Physics Phenomenology after the Higgs Discovery*, and through Germany's Excellence Strategy EXC 2181/1 – 390900948 (the *Heidelberg STRUCTURES Excellence Cluster*). TH was funded by the Carl-Zeiss-Stiftung through the project *Model-Based AI: Physical Models and Deep Learning for Imaging and Cancer Treatment*. The authors acknowledge support by the state of Baden-Württemberg through bwHPC and the German Research Foundation (DFG) through grant no INST 39/963-1 FUGG (bwForCluster NEMO).

## A    2-dimensional correlations

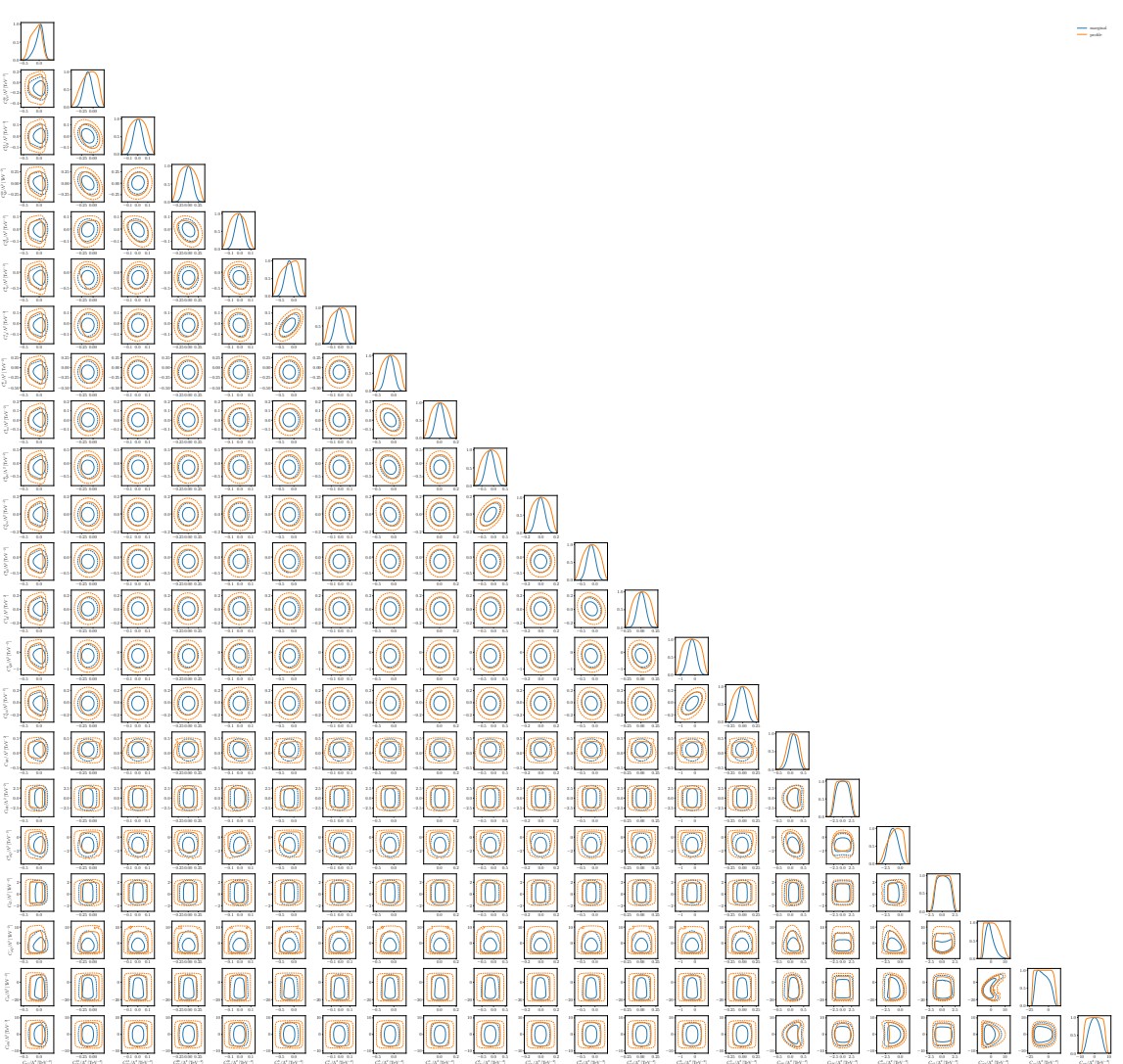

Figure 6: 2-dimensional and 1-dimensional profile and marginalized likelihoods for the top sector.

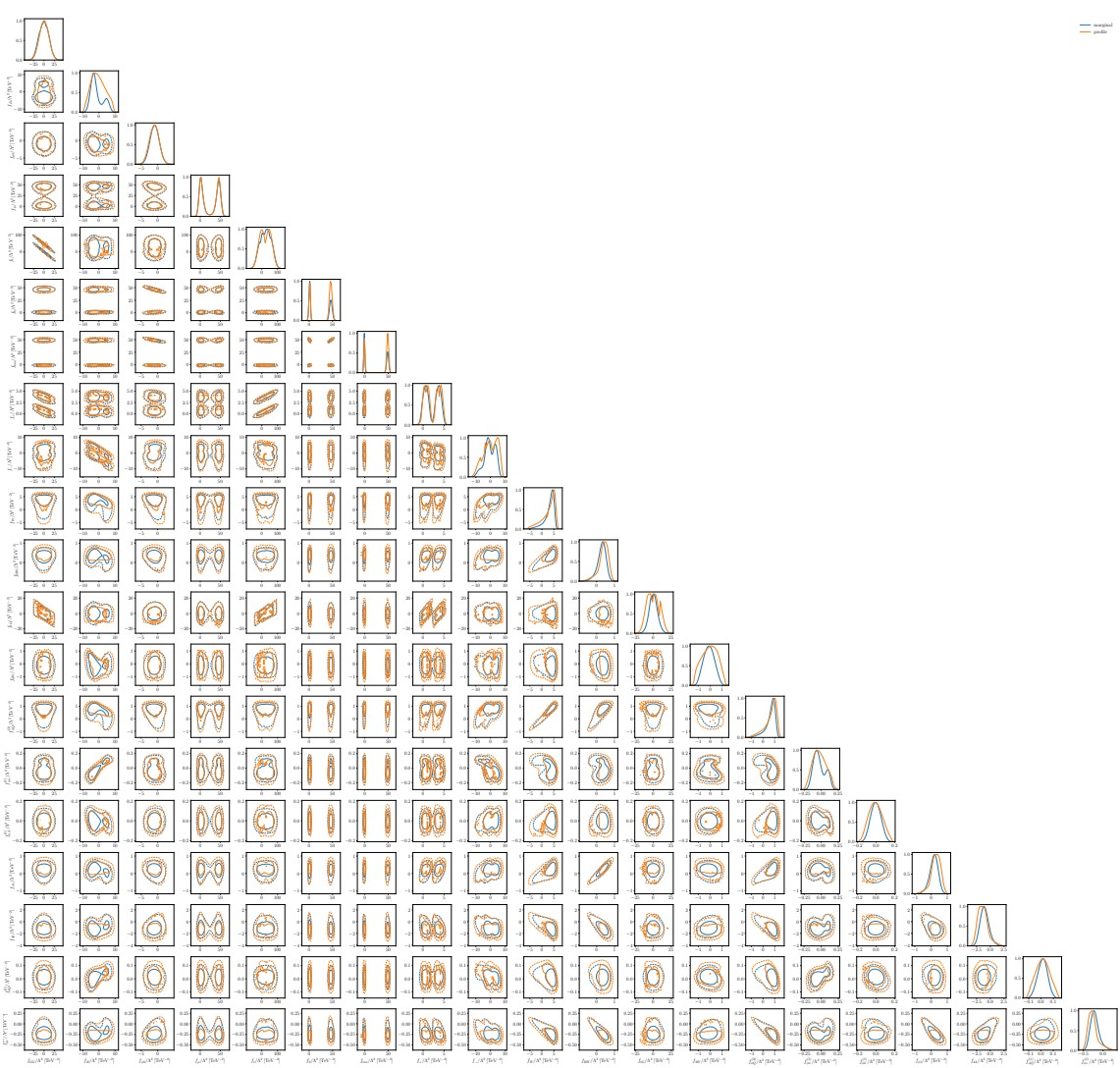

Figure 7: 2-dimensional and 1-dimensional profile and marginalized likelihoods for the Higgs-gauge sector in the HISZ operator basis.

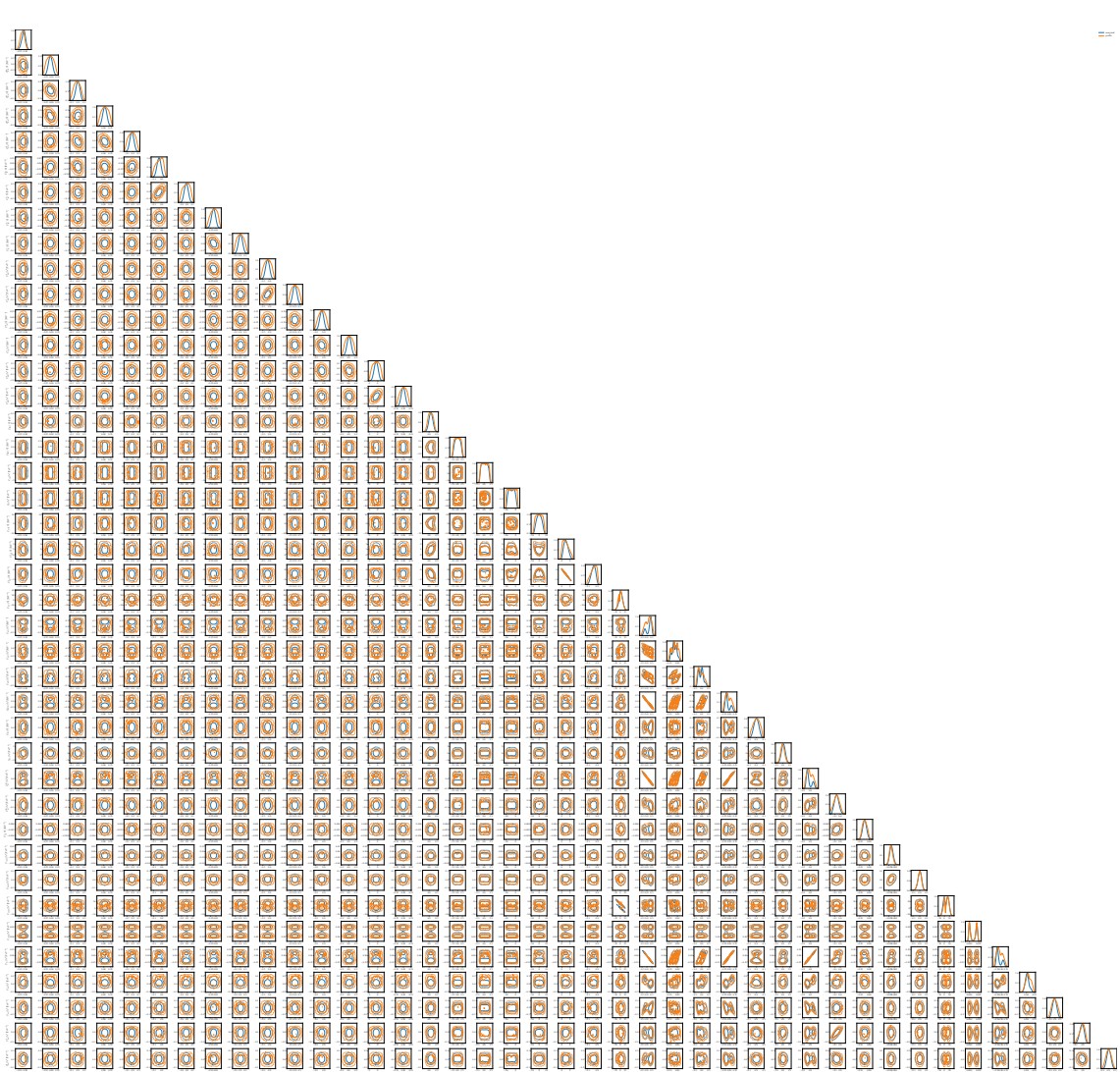

Figure 8: 2-dimensional and 1-dimensional profile and marginalized likelihoods for the Higgs-gauge and top sectors combined.

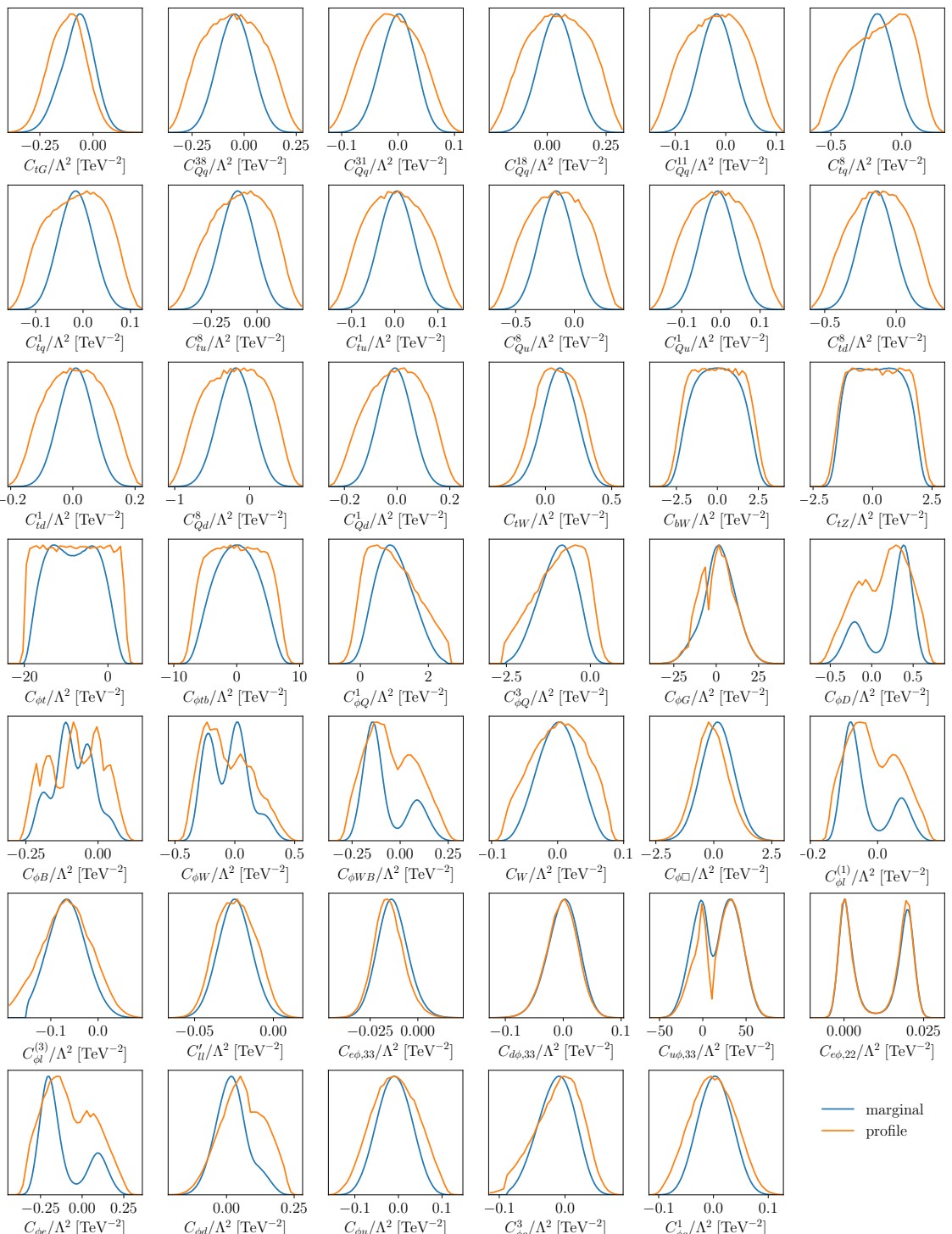

Figure 9: 1D profiled (orange) and marginalized (blue) likelihoods for the combined fit.

# B  Hyperparameters

|  |  | Top | Higgs-gauge | Combined |
|---|---|---|---|---|
| Architecture | Coupling blocks | RQ splines | | |
| | Spline bins | 16 | | |
| | Subnet layers | 3 | | |
| | Hidden layers | 64 | | |
| Pre-scaling | Number of samples | 10240 | 40960 | 40960 |
| | AIS steps | 1500 | 5500 | 5500 |
| | Target acceptance | 0.33 | | |
| Pre-training | Batch size | 1024 | | |
| | Epochs | 15 | 6 | 6 |
| | MCMC steps between batches | 20 | 10 | 10 |
| Training | Learning rate | 0.001 | | |
| | Batch size | 1024 | | |
| | Batches | 100 | 2000 | 6000 |
| | AIS steps | 4 | 4 | 8 |
| | Buffer capacity | 262k | | |
| | Ratio buffered/online steps | 6 | | |
| Sampling | Batches | 100 | 2000 | 1000 |
| | Batch size | 100k | | |
| | Marginalization bins, 1D | 80 | | |
| | Marginalization bins, 2D | 40 | | |
| | Profiling bins, 1D | 40 | | |
| | Profiling bins, 2D | 30 | 30 | 20 |
| Profiling | Batch size | 100k | | |
| | Optimizer | LBFGS | | |
| | Optimization steps | 200 | | |

Table 5: Hyperparameters for the five fitting steps. If only one value is given, it applies to all three fits.

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
