# Peer review of "Profile Likelihoods on ML-Steroids"

_SciPost Physics_

## Round 2 · Referee Report · Andrew Fowlie (Referee 1) · 2025-6-24

Strengths

  1. Considers computational method for profile likelihood - this topic is arguably neglected relative to sampling from posterior
  2. Novel method that exploits GPU and developments in ML 3.Appears to be extremely performant in practice; allows SMEFT analyses that were previously infeasible

Weaknesses

  1. Lack of theory for assessing performance of profile likelihood algorithms
  2. Typos and errors in equations
  3. Proposed method is a complicated combination of existing techniques
  4. Limited discussion & comparisons to existing state of the art

Report

I have spent a few hours reading and thinking about the paper. The paper introduces a new technique (or combination of techniques) for computing the profile likelihood in the context of SMEFT. The paper is on the whole a useful and relevant contribution and I recommend publication after my comments have been addressed.

Introduction

The introduction would be improved by clarifying the general problem, the definition of the profile likelihood, motivations for computing it, and existing state of the art. E.g., what other methods have been used in the past? Some details of the problem at hand would also be useful: how big is the parameter space? how costly is an evaluation of the likelihood function? are the targets pathological, e.g., multimodal or degenerate?

The five steps to happiness

The second section introduces the setting and techniques. Unfortunately, the equations contain typos:

  • eq. 1: Why isn't the LHS a function of the parameters $c$?
  • eq. 2: The normal distribution is characterized by a mean and a variance; only one of them is indicated in eq. 2
  • eq. 3: The LHS is a function of $\theta$; however, those parameters are profiled on the RHS. The meaning of $\mathcal{F}$ isn't clear: I would guess it's flat and centered at zero, but what precisely is $\sigma$ in this case? The standard deviation? or is it flat between $-\sigma$ and $+\sigma$?
  • eq. 4: Normalization constants have been dropped on the RHS (e.g., $\log \sqrt{2\pi\sigma^2}$ terms). This might be of no consquence, but it should be made clearer
  • eq. 5: The notation $\text{Poiss}(\tilde s | d, bk)$ was confusing to me: I guess this means we should consider it as a function of $\tilde s$, but it isn't a Poisson distribution in $\tilde s$

There was an approximation that I didn't understand, eq. 9. If I understand it, we are approximating the product of likelihoods by their reciprocal sum. Why are we doing this? Why can't we evaluate the product? What terms are being neglected? The equation is strange since one of the terms, 1 / gaussian, is a density and is dimensionful. Comparing to eq. A2 of [https://arxiv.org/pdf/0904.3866], perhaps there are typos here as well.

Later, the authors clarify, 'sampling alone is not sufficient to get smooth profile-likelihoods, so it has to be combined with a maximization procedure'. I'm unsure about what is meant by sampling here and throughout - the likelihood isn't a distribution in the parameters.

This is the first mention of 'smooth profile-likelihoods'. Why do we desire them? If we just want to compute confidence intervals via Wilks' theorem, we just need particular level-sets of profile-likelihood. If we really want a plot to look smooth, a brief discussion of estimation methods (e.g., analagous to density estimation - kernel density estimation or histogramming) would be helpful.

Finally, the new procedure is introduced. The procedure consists of several rounds: a traditional sampling method, such as MCMC, followed by normalizing flows and neural importance sampling, and finally polished with traditional optimization algorithms. This fact that it is quite complicated makes me concerned that it might not generalize well. Nevertheless, as we shall see it appears to work very well.

SMEFT & Results

The results focus on the computational method, rather than novel physics insights. To start with examples show that a spiral function was wonderfully resolved by the new method, in both the profile and marginal. We continue to physics cases, and see that the new method allows a complete fit in a few hours versus about 24 hours previously.

I found it hard to understand the comparison: what is the metric for performance of a profile likelihood evaluation? From table 4, we see effective number of samples. This measure quality of posterior expectations from a sample based representation of the posterior, since errors $\propto 1 / \sqrt{n_\text{eff}}$. What does it mean in the context of profile likelihood? I think the discussion of the comparison metrics could be improved, even if it was just that there was limited theory in this setting.

Nevertheless, even without any theory to characterise performance here, it is clear from e.g. '8200 for a single CPU vs a single GPU. A complete set of profile likelihoods were, essentially, out of numerical reach for the current CPU implementation' that the new method works well and allows things that were previously impossible.

Requested changes

See report.

Typos

  • this analysis is possibly, but extremely CPU-intensive => possible

Recommendation

Ask for minor revision

---

## Round 2 · Referee Report · Humberto Reyes-González (Referee 2) · 2025-7-13

Strengths

-Use of well chosen novel techniques for a relevant problem, achieving orders of magnitude improvements.
-Well written introductory chapter describing how to construct profile likelihoods, as well as the particular challenges to be tackled.
-Novel approach tested on both toy and realistic examples demonstrating the efficiency of the novel methodology

Weaknesses

-Minor clarifications required. - Use of supplementary performance metrics recommended.

Report

The manuscript describes a considerable upgrade to the methodology used for sampling profile likelihoods from global SMEFT analyses within the dedicated SFitter framework. Approximating likelihood functions in global SMEFT analyses is a complex, high-dimensional problem. Improvements such as the one presented in the manuscript are welcome, as they potentially allow for more efficient studies of the Standard Model and beyond at the LHC. The authors implement modern techniques such as neural importance sampling and normalizing flow–based priors for MCMC, together with GPU-based parallelized computing, to achieve orders of magnitude faster sampling of profile likelihoods compared to previous CPU-based methods. The submission meets the criteria to be published on SciPost Physics. Results are presented for a number of benchmark cases. Before giving my recommendation for publication, I kindly ask the authors to address the following comments.

Requested changes

  1. What exactly is meant by 1D and 2D marginal distributions? Are these simply rankings of values over certain dimensions, or do they refer to integrating out nuisance parameters? A brief clarification would enhance the interpretability of the figures. 
2. It is mentioned that the new method produces smoother profile likelihoods. An explicit explanation (or example) of the advantage of this smoothness would strengthen the statement. Additionally, if the difference is visually noticeable, a figure comparing CPU-based vs. GPU-based profiling would be helpful.

3. GPU parallelization leads to faster sampling. However, if I understand correctly, the use of NIS and NFs should also enable more efficient convergence and better coverage of the MCMC walks. Supplementary convergence diagnostics would support these claims and offer further assurance that the maxima have been properly found and sampled. These could include Geweke scores, KS (or chi-squared) tests within and between chains, and Gelman–Rubin statistics. They would be a valuable addition to the ESS metrics already presented in the paper.

  1. For my own understanding: in Fig. 2, the different maxima in the 1D plots are well reproduced by the profiled likelihoods. This is not the case in Fig. 1 (granted, the 1D marginal is more complex). Why is this the case? When I look at the 2D plot in Fig. 1, it appears to be quite accurate.

  2. For reproducibility purposes, a reference to the code and/or data should be provided if available.

  3. There is a typo in the last line of the first paragraph in the Introduction. both sectors combined [10–13] -> and both sectors combined [10–13].

Recommendation

Ask for minor revision

---

## Round 2 · Referee Report · Anonymous (Referee 3) · 2025-7-25

Report

This paper describes an efficient technique for inferring statistical information about large numbers of fundamental theoretical parameters based on experimental data. This is applied to determining confidence intervals for a large set of SMEFT parameters based on LHC data. Their results in principle contain all correlations between SMEFT coefficients, and some example plots are shown demonstrating these.

The innovation in this paper is the use of sophisticated sampling techniques to be able to extract samples from the posterior distribution based on a likelihood function. The combination of these techniques, which consist of "five steps to happiness" and include annealed sampling and normalizing flows in MadNIS, are state-of-the-art. They are well described, novel and interesting. The application to the problem at hand is also interesting and relevant. I therefore recommend the paper for publication in SciPost.

When reading the paper, my only disappointment was that it was hard or not possible to find clear references to the actual experimental data that is taken into account. For example, is it only LHC data, or also LEP (for the EWPOs?). I would appreciate some additional information or clearer references to which SFITTER references explain exactly what goes into their likelihood.

Recommendation

Ask for minor revision

---

## Editorial Decision

awaiting_resubmission